# Application of Fluorescence In Situ Hybridization (FISH) in Oral Microbial Detection

**DOI:** 10.3390/pathogens11121450

**Published:** 2022-12-01

**Authors:** Junjie Gu, Huayu Wang, Mengye Zhang, Yichen Xiong, Lei Yang, Biao Ren, Ruijie Huang

**Affiliations:** 1State Key Laboratory of Oral Diseases, National Clinical Research Center for Oral Diseases, West China School of Stomatology, Sichuan University, Chengdu 610041, China; 2Department of Pediatric Dentistry, West China Hospital of Stomatology, Sichuan University, Chengdu 610041, China

**Keywords:** fluorescence in situ hybridization, microbial detection, oral microecology, oral microorganism

## Abstract

Varieties of microorganisms reside in the oral cavity contributing to the occurrence and development of microbes associated with oral diseases; however, the distribution and in situ abundance in the biofilm are still unclear. In order to promote the understanding of the ecosystem of oral microbiota and the diagnosis of oral diseases, it is necessary to monitor and compare the oral microorganisms from different niches of the oral cavity in situ. The fluorescence in situ hybridization (FISH) has proven to be a powerful tool for representing the status of oral microorganisms in the oral cavity. FISH is one of the most routinely used cytochemical techniques for genetic detection, identification, and localization by a fluorescently labeled nucleic acid probe, which can hybridize with targeted nucleic acid sequences. It has the advantages of rapidity, safety, high sensitivity, and specificity. FISH allows the identification and quantification of different oral microorganisms simultaneously. It can also visualize microorganisms by combining with other molecular biology technologies to represent the distribution of each microbial community in the oral biofilm. In this review, we summarized and discussed the development of FISH technology and the application of FISH in oral disease diagnosis and oral ecosystem research, highlighted its advantages in oral microbiology, listed the existing problems, and provided suggestions for future development..

## 1. Introduction

The human oral microbiota consists of more than 700 species, and the microbial communities are diverse at different niches in the oral cavity [1]. Human oral microbiota mainly consists of bacteria, fungi, viruses, protists, and oral archaea [2,3]. According to the human oral microbiome database (HOMD), 70% of the oral bacteria are culturable, 30% are unculturable, and only 57% of culturable species have been named [4]. 

Human oral microbiota has distinct characteristics [5,6]. The human oral cavity provides an unexplored, structurally complex environment and a differentiated set of habitats [7], and saliva promotes the interactions of various local environments [8]. The spatial organization of microorganisms in the oral cavity is formed by dynamic equilibrium flow and adhesion, abscission, and colonization, as well as the interaction between microorganisms and host [7,8], which makes oral microbiota an important biological model of the human microbiome [9]. The synergies and interactions of variable oral microorganisms help the human body to maintain oral health, but the imbalance of oral microbiota leads to the development of oral diseases, even systemic diseases [10], indicating that monitoring of oral microbiota is important for the diagnosis or analysis of oral diseases and some systemic diseases [11]. 

Various oral microbiological detection methods have been developed in labs and clinics [12]. Fluorescence in situ hybridization (FISH), a non-radioactive technique combining cytogenetics and molecular biology, was developed based on radiogenic in situ hybridization in the late 1980s to replace isotope markers with fluorescent markers [13]. The application of FISH in microbial detection usually targets microbial ribosomal RNA (rRNA) genes [14]. FISH can also be used to detect the effects of different external factors on colonies and the interactions between colonies [15]. In present review, we have summarized and discussed the applications of FISH in oral microbiology to highlight its advantages in oral microbial detection and oral diseases diagnosis (Figure 1).

## 2. Research Progress of FISH

FISH is one of the most routinely used cytochemical techniques for genetic detection, identification, and localization by a fluorescently labeled nucleic acid probe. It is more specifically used in hybridizing processes with nucleic acid sequences of interest [16]. The vitality of this technology is continuously proved by the evolution of probe design, signal amplification, and multiplexing, broadening the application of experimental research and clinical diagnoses in this field of study [17]. Specifically, FISH and its variants have been applied to nucleic acid investigation, cell metabolic research, oncology diagnostics, and microbiological research [18].

### 2.1. Development of FISH

In the 1960s before the advent of FISH, cytochemical methods of detecting and localizing specific intracellular molecules were mainly immunocytochemistry based on in situ hybridization with radiolabeled probes and antigen-antibody interaction with fluorescein-labelled immunoglobulins [19,20,21]. However, radioactive RNA or DNA binding to DNA sequences in situ used to be limited to detecting, characterizing, and localizing specific DNA segments. Due to the low resolving power and long exposure time in autoradiography, this methodology is inaccurate in quantification [17,22]. To compensate for this deficiency, indirect immunofluorescence, a type of immunocytochemistry technique developed in 1965, has been successfully applied in quantitatively analyze biological markers via fluorophore-labeled immunoglobulins that target marker-binding antibodies [21]. 

As a combination of in situ hybridization and indirect immunofluorescence, a method described by Rudkin et al. in 1977 replaced autoradiography with fluorescence microscopy to detect in situ signals from antibodies against DNA-RNA hybrids [23]. Before long, based on the methods above of nucleic acid hybridization and indirect immunofluorescence without intermediated antibodies, Bauman et al. first applied fluorophore-covalent-labeled RNA probes in specific DNA sequence detection in 1980, representing the birth of FISH [22]. Nonetheless, the application of this original FISH still needs further improvement in probe affinity and signal amplification due to the low signal intensity caused by target inaccessibility and low copy numbers [24].

The optimization of probe design and synthesis is crucial to the development of FISH. The establishment of genetic databases made it possible to design probe sequences targeting complementary gene sequences of interest [25]. Probe preparation underwent a development from manual multiple sequence alignment for conserved target regions to probe sequence auto-selecting programs and even web-based platforms for theoretical evaluation of the probe performances in FISH [25,26,27,28,29]. The new technology makes it easier to control the length, complementary sequence, thermodynamic property, potential secondary structure, and specificity of the probe, which are directly related to the successful application of FISH [30]. 

With accurate probe design methods, FISH properties became even more modern, with changes that include increasing signal intensity and stability, range of targets, and sensitivity. Commonly used variants are: catalyzed reporter deposition FISH (CARD-FISH), gene-FISH, recognition of individual genes FISH (RING-FISH), nucleic acid mimics FISH (NAM-FISH), combinatorial labeling, and spectral imaging FISH (CLASI-FISH), double labeling of oligonucleotide probes for FISH (DOPE-FISH), as described below. 

The polynucleotide probe also has an advantage in signal intensity with multiple labels and secondary structures mediating probes-connected networks [31]. RING-FISH achieves high detection efficiency to single genes with multiple labeled transcript polynucleotide probes generating halo-like signals [32]. As an alternative to RING-FISH for prokaryotic cellular microorganisms, two-pass TSA-FISH provides higher efficiency and a better signal-to-noise ratio, making it a useful protocol based on the functional gene for single microbial cell detection [33]. By combining rRNA CARD-FISH and polynucleotide probe gene detection, gene-FISH can be an innovative technique to provide a stable signal and high sensitivity [34]. Furthermore, direct gene-FISH for gene signal quantification can be formed by replacing CARD (and its disabilities) with fluorochrome-labeled probes and super-resolution microscopy [35]. 

The nucleic acid mimics probe has become the most promisingly high-efficient probe with higher affinity, specificity, and better stability based on resistance to enzymatic degradation [36], used to overcome the issues with weak signal caused by low affinity and vulnerability of DNA or RNA probes. The NAM probes currently being utilized are peptide nucleic acid (PNA), 2’-O-Methyl-RNAs, and locked nucleic acid (LNA) [37]. 

The variants mentioned above show corresponding advantages in detecting microorganisms, allowing direct identification without cultivation and further detection of microbial community structure and individual function [36,38]. As microbiological research kept expanding from individuals to communities, multiplexing identification was developed to simultaneously analyze multi-microbes and microbiota efficiently. A typical example is CLASI-FISH, which introduces combinatorial labeling and spectral imaging (CLASI) into FISH and can be applied to distinguish several microbes at once by linear unmixing the spectra of fluorophores from overlapping spectra [39]. However, CLASI-FISH application may be limited due to internal sensitivity loss and potential probe binding bias caused by binary combinations, limiting the application of this technique for quantification analyses [40]. Compared to CLASI-FISH, DOPE-FISH provides a double signal intensity as well as stable specificity and higher affinity to targets, but the detectable number of microbes is much lower [41].

The variants of FISH are improving, as are their applications based on the detection of nucleic acids. Apart from microbiological applications, FISH has been a gold standard technique in absolute gene copy number quantification in cancer, allowing implementation of precise treatment strategies [42]. Based on imaging spatial transcriptomics, FISH performs cell segmentation for cell interactions and the state of complex tissues for analysis of, for example cell organization in the cerebral cortex and cell fate decisions in organogenesis [43,44,45]. Expansion-Assisted Iterative (EASI)-FISH was developed for the 3D organization of cell types in thick tissue, contributing to characterizing the architecture underpinning brain function [46]. It is reported that the combination of FISH and small and ultrabright fluorescent polymeric nanoparticles functionalized with DNA allows a simpler, faster and sensitive single-cell RNA imaging method for transcriptomic analysis [47]. The characteristic chromosomal abnormalities in cancer cells can be obtained by high-resolution karyotyping by FISH, which can help in differential diagnosis [48]. Other uses of FISH karyotype analysis include genetic diagnosis, prenatal screening, and plant and animal genetic studies [48,49,50,51,52]. But karyotyping is highly dependent on frequency of cytogenetically abnormal cells, in need of enrichment methods such as fluorescence-activated cell sorter (FACS) to improve the sensitivity [53]. Thus, the combination of FISH with other suitable techniques can compensate for some inherent drawbacks and break the application limitations of FISH. The coupling of Flow Cytometry and FISH enables high-throughput quantification of complex whole-cell populations, and the association with FACS (FLOW-FISH-FACS) enables sorting of target microorganisms [54]. In drug discovery, FISH was introduced into high-content screening for intracellular imaging of mRNA to screen mRNA-associated drugs and assess their pharmacological activity [55]. Resolution After Single-strand Exonuclease Resection (RASER)-FISH provides a robust generation of single-stranded DNA with excellent preservation of chromatin structure, nuclear integrity and improved hybridization efficiency, which is achieved by exonuclease digestion rather than physical denaturation by heat and exposure to formamide [56].

### 2.2. Procedures and Principles of FISH and Its Variants Used for Oral Microbial Detection

The general protocols of FISH include (a) specimen treatment, (b) probe denaturation, (c) hybridization, (d) elution, (e) hybridization signal amplification (applicable to biotin-labeled probes), (f) re-staining, (g) encapsulation, and (h) fluorescence microscope observation of FISH results [40,57,58]. Probe labeling and specimen processing require different processing methods according to different detection needs, in order to obtain clearer detection results in the subsequent hybridization process [59,60]. For example, to substantially enlarge the number of distinct taxa in one FISH experiment, CLASI-FISH was created, and the established FISH protocols are ordinarily suitable for the hybridization, but still some changes for the protocol were proposed, such as fixing samples with PFA followed by ethylalcohol [39]. Take CARD-FISH as another instance, to enhance sensitivity and whittle background interference, the protocol resembled the typical FISH, but tyramide-fluorophore and HRP was introduced to replace fluorophore on the probe [61]. All the protocols are detailedly elaborated by the inventors of the FISH variants, so that researchers can consult relevant papers according to application needs. Different physical and chemical environments during hybridization will affect the effect of hybridization, and redyeing will significantly change the clarity of the observed objects, which will lead to significantly different results under the optical microscope [62,63].

FISH can evaluate various types of test samples, such as tumor tissue, pathological sections, local animal samples, human tissue samples, etc. For tissue samples: 4% paraformaldehyde fixation (paraffin section), or put into liquid nitrogen, −80 °C storage, sample not less than 100 mg; For cell samples: cell slides prepared with 6-well plates, with no less than 2 × 10^6^ cells per well; For environmental bacteria: sludge samples or sludge particles and other sample forms that can be used for smear or sectioning [6,39,64,65]. 

Though FISH is widely used in microbiological detection and biofilm analysis, some techniques belonging to nucleic acid, immunity, and single-cell techniques also have been applied in different emphases of microbial research due to their different advantages. Ten common techniques (including FISH) in microbial research were selected and compared based on several performance factors, including detectable resolution, applications in oral microbiology, culture reliance, quantification appliance, capability for microbiota analysis, and unsearched species (Table 1). In summary, some features of FISH make it an advantageous technique, e.g., precision, culture-independent, unsearched species detectability for microbiota analysis for extra semi-quantification and intuitive spectral imaging.

Furthermore, most oral microorganisms are unculturable, and the community structure and components are complex and changeable, and these properties make FISH very suitable for oral microbial research. Among many variants of FISH, classical FISH and CLASI-FISH are the most popular in terms of their application in oral microbiology. However, it is noteworthy that a new variant called HiPR-FISH is considered to have a promising future. The procedure and principles of these three currently used FISH techniques in oral microbiology will be elaborated throughout this study.

#### 2.2.1. FISH

The most common procedure followed by researchers who apply FISH involves the following four steps: (1) fixation for dehydration to inhibit the action of enzymes without cell and nucleic acid structures destruction (some special cells need extra permeabilization treatment that degrade the cell walls to increase membrane permeability); (2) hybridization between target nucleic acid sequences and specific complementary probes labeled with fluorescent dyes or reporter molecules to be detected by fluorescent antibodies; (3) washing to remove the unbound or loosely bound (usually mismatched) probes; (4) detection and visualization of the probe-bounded cells by typically fluorescence microscopy and some advanced microscopy systems, then access to information for species detection or microbiota analysis [16,36,87]. Hybridization is one of the most decisive steps, strongly influenced by probe properties on specificity and sensitivity [87]. These four steps are also the basic standpoints of modified FISH variants.

#### 2.2.2. CLASI-FISH

Regardless of some minor adaptions to classical FISH protocols for CLASI-FISH application, the main improvement of CLASI-FISH are probe design and image acquisition/analysis. CLASI-FISH probes are particularly effective in synthesizing two probes for one targeted sequence and two different sequence-particular fluorophores; thus, the cells of different species are labeled by specific combinations of two highest-intensity fluorophores [88]. For detection, CLASI-FISH applies confocal laser scanning microscopy (CLSM) in image acquisition, allowing the linear spectral unmixing computational analysis to identify the fluorophores with overlapping spectra and analyze the fluorophore composition in each cell [39].

#### 2.2.3. HiPR-FISH

Regarding the expansion of the number of distinguishable species in a single image as CLASI-FISH, it has been previously verified that High Phylogenetic Resolution FISH (HiPR-FISH) achieves significantly higher taxonomic resolution and multiplexity due to a two-step hybrid approach and a routine for automated image segmentation [89]. Two kinds of probes were used for hybridization: the encoding probe in the first step and the readout probe in the second step. The encoding probes are taxon-specific probes that include targeting sequences modified with different flanking readout sequences. The fluorescently labeled readout probes target readout sequences and stochastically bind to the bound encoding probes, representing equal proportions of fluorophores. 

Ten distinct fluorophores encode up to 1023 fluorophore combinations, and one species corresponds to a 10-bit binary barcode derived from designed one or more encoding probes binding with relevant readout probes and presenting different spectral components. After image acquisition, automated image segmentation classifies the spectra of images and assigned cells the corresponding barcodes. The reference spectrum for each barcode is established by the Förster resonance energy transfer (FRET) model, and the k-means clustering-denoised and straightened images of cultured single-cell and biofilm samples are segmented by the watershed algorithm that the seed is defined by developed Local Neighborhood Enhancement (LNE). For spatial analysis, the adjacency segmentation is also generated by the watershed algorithm and then calculates the intuitive spatial association matrix. The super-resolution images and 3D datasets are generated the same way, and the 4D data cube needs rendering in ipyvolume.

## 3. Research and Application of FISH in Oral Microbial Detection

FISH has been widely used in various fields of oral microbial research. From the basic research at the practical level to the potential clinical application and outcome transformation, the important role of this methodology has been witnessed by scientists in different scenarios inside the research community. In this section, we summarize the three major applications of FISH and explain its relevant details (Table 2).

### 3.1. Diagnosis Assistance

This function of FISH has been used to identify the species of pathogenic microorganisms in the simplest way, but it is still in the basic research stage in terms of predicting the risk of disease and prognosis, which needs to be further tested and verified in clinical practice. In the context of modern medicine, the requirements of patients and clinicians for diagnosis have upgraded from merely knowing the type of diseases to detecting specific causes and predicting disease progression. For patients with diseases related to oral microorganisms (caries, pulpitis, periodontal disease, etc.), most of them exerted evident symptoms at the time of its’ initial diagnosis, such as pain, inflammation, and changes in biological characteristics. And increasing studies have shown that cancer, atherosclerosis, diabetes and other diseases can be reflected in oral flora. Oral microorganisms play an important role in the occurrence and development of these diseases, and the changes in community structure can somehow indicate how it has been progressing [110]. Therefore, we can understand the types of pathogens based on determining the types of diseases and improve the accuracy of diagnosis to outline potential target therapies. At present, most of the detection methods of pathogenic microorganisms in hospitals are still traditional identification methods, including Gram staining, microbial culture and biochemical tests. However, for the complexity of oral microenvironment, traditional methods are not well qualified for multi-species identification, community analysis and other aspects, let alone judging the risk of disease and prognosis. As a highly-stable, high-resolution, simple, and intuitive nucleic acid probe technology, FISH can effectively facilitate the application of such functions. However, more detailed databases are still needed to support accurate diagnosis due to inadequate personalized research and global data sets.

Clinically, FISH can be used to detect and analyze various disease-causing factors, such as HIV and Epstein-Barr virus, providing a faster way to diagnose such conditions. However, due to the spreading of the new COVID-19 pandemic, nucleic acid testing has become a part of people’s lives, so we urgently need a more rapid and more convenient way to detect the virus. Hepp et al. developed a rapid FISH protocol capable of quantitatively detecting influenza virus, avian infectious bronchitis virus, and SARS-CoV-2 in approximately 20 minutes by combining nasal and throat swabs with the added virus in virus cultures. This rapid and simple method can be used both as a commonly used detection technique in the laboratory and to aid the future diagnosis of enveloped viruses with accessible genomes [62]. 

Modern research shows considerable differences in the type, proportion, and structure of oral microbiota between healthy and pathological states, which can be used as breakthrough points to diagnose the actual causes of the disease. FISH was initially used to detect relatively single and known pathogenic microorganisms, but with the development of new technology, a variety of microorganisms can now be detected simultaneously. Clinicians and researchers use FISH to detect microbes’ information, such as species, and community structure, after extracting samples from patients. This kind of data is included in the databases and also used as feedback to clinicians to develop more accurate treatment plans. Taking oral lichen planus (OLP) as an example, there are significant differences in microflora between OLP patients and healthy controls. By applying FISH, Zheng et al. [91] and Wang et al. [92] revealed species, location, and average optical density (AOD) variations of oral microorganisms between OLP patients and healthy individuals. These significant differences might all become targets for diagnosis and treatment in the future. 

In addition to being used in the diagnosis of diseases, the results of FISH can be used in the formulation of treatment options. Take the most common and inconvenient form of bad breath as an example. Bernardi et al. visualized and quantified the dorsal biofilm through the combination of FISH and CLSM and stained eubacteria, *Streptococcus* spp., and *F. nucleatum* using specific fluorescent probes. In their experiment, they found a significantly higher proportion of *F. nucleatum* and *Streptococcus* spp. in the biofilm of the halitosis group. It was concluded that the relative ratio of total microorganisms to these two bacteria could be considered a relevant factor in causing bad breath; hence it can be included in the treatment goals [93].

Apart from oral diseases, FISH is also used for microbial detection of lesions of other diseases. For example, it has been found that there are many common oral bacteria in oral squamous cell carcinoma [111]. Although oral microorganisms are also found in atherosclerotic plaques [112] and Alzheimer’s disease patients’ brains [113], the detection method used is PCR. Therefore, FISH is almost equal to PCR in determining whether there is a certain bacterium or not. Nevertheless, considering that FISH technology does not depend on the isolation and culture of bacteria and that not all laboratories have the experimental conditions for FISH, the specific method to choose eventually depends on the experimenters. However, if we want to expand the detection scope to more species, community distribution, and even predicting risk and prognosis, FISH will be more advantageous.

### 3.2. Oral Microecology

Currently, research focus on oral microecology occupies an important position in stomatology [114], and many past research results showed that oral microbes remain a major cause of a variety of oral diseases [63]. Many contemporary studies indicate that FISH is a powerful tool for studying oral microecology [115]. In recent years, due to the rapid development of this technique, it has become a highly automated process with two major functions, microbial species identification and community monitoring [116]. Furthermore, with the continuous combination with other molecular biology techniques, it has been shown that FISH can even be used to detect the effects of different external factors on colonies and interactions between them [15]. 

According to the Expanded Human Oral Microbiome Database (eHOMD), only 57% of oral microorganisms can be cultivated and formally named, 13% of oral bacteria can be cultivated but not named, and 30% of the oral microorganisms fail to be separated from the biofilm, hence hindering the study of oral micro-ecological research [117]. FISH, which does not require bacteria culture, can identify microbial populations into genera and species by using fluorescently labeled specific oligonucleotide fragments as probes to hybridize with DNA molecules in the environmental genome. This feature helps to avoid the limitations of traditional culture methods for identification and counting. It has high application value in identifying oral microflora, bringing a new detection method for some diseases caused by bacterial infection. 

Annett et al. detected the presence of Treponema pallidum in the tissue samples of syphilis patients by a specific 16S rRNA FISH probe against T. pallidum. Their effective methodology avoided possible false results of serological identification in HIV-positive patients and provided a faster and more effective technique for disease monitoring [90]. Using similar methods, Fernandes et al. identified bacteria in pulp infections by FISH and confirmed the involvement of bacteria such as Pyramidobacter piscolens and Fretibacterium fastidiosum. These two bacteria are also challenging to detect in endodontic infections because they are difficult to separate and culture [96]. In addition, FISH can also visualize oral colonies with a confocal laser scanning microscope (CLSM), thus locating microorganisms’ data in clinical samples and assisting in speculating the process of invasion, colonization, and transmission. Bertl et al. successfully monitored the position of the oxygenic pathogens in the biofilm of the vocal cord prosthesis by the combination of FISH and CLSM, supporting that they were not the cause of the breakdown of vocal cord prosthesis equipment [97]. In contrast, in Zangeneh et al.’s experiment, they used PCR-DGGE to detect only the type and number of bacteria in the oral cavity of multiple sclerosis patients [79]. This shows that compared to traditional microbial detection techniques such as PCR, FISH not only has the advantage of being a simple and fast process, but by combining it with other detection methods, more information about the colonies can be obtained.

Due to its ability to display location information apart from identifying species, researchers have increasingly applied FISH to monitor microbial communities in recent years. The technique can be used to study colonies that are difficult to cultivate and clearly present the overall microbial environment without requiring additional bias-prone steps such as extraction and amplification. FISH can accurately and quickly reflect the in-situ distribution of microbial communities in various systems, avoiding the complex process of other community analysis methods [118]. Through FISH, we can quickly and efficiently analyze the microbial communities in pathological conditions and then compare them with the normal ones to assist doctors with disease diagnosis.

Bring et al. spotted the distribution of TM7 bacteria, which is difficult to culture in the oral microecology through FISH and further reveals the mechanism of their metastatic infection [59]. Shi et al. created a micron-scale map of the location and identity of hundreds of microbial species in a complex microbial community by high-phylogenetic-resolution microbiome mapping by fluorescence in situ hybridization (HiPR-FISH), based on spectral imaging and decoding. Their findings revealed the disruption of the spatial network in the mouse gut microbiota by antibiotic treatment and the longitudinal stability of the spatial structure in the oral plaque microbiota, providing a framework for the spatial analysis of microbial communities at single-cell resolution [89]. By testing the oral microbial community at different times, FISH plays an essential role in static detection; it dynamically observes the changes of the microbial community in the mouth and monitors the occurrence and progression of the disease. 

Using the combination of FISH and CLSM, AI-Ahmad et al. measured microflora changes in oral plaque within eight weeks with a particular focus on individuals with poor oral hygiene [57]. Combining with other different molecular biological technologies, the quantity, location, and morphological changes of different microbial communities in the biofilm could be dynamically observed, which further yields the influence of various factors on specific bacteria and the interaction between microbial communities. Similarly, Esteves et al. used FISH to detect the interactions of different microbial communities in dental plaque, revealing the impact of these interactions on the generation and development of paradentosis [98]. Moreover, FISH combined with flow cytometry for detecting microbes is a promising technology to diagnose and evaluate microbial community structure and its dynamics. Its highly automated operation makes it more suitable than other methods for frequent and rapid monitoring of specific colonies [119]. 

Because of its two effective features for species identification and community detection, FISH is often used to study dental plaque biofilm. Mature dental plaque consists of multiple species of biofilms and contains more than 500 different bacteria, causing common oral diseases such as periodontitis, dental caries, and gingivitis [120]. Combined with CLSM, this technique allows the visualization of biofilm systems, and the dynamic monitoring of plaque biofilms can assist doctors in the judgment of disease progression while avoiding the traditional post-culture detection, identification, and quantification. For instance, Gmür et al. identified and analyzed single microbial cells in the dental plaque through immunofluorescence and FISH [94]. Karygianni et al. found that combining CLSM and FISH could create high-resolution 3D images of individual bacteria in their natural environment to better visualize bacterial communities in dental plaque, allowing professionals to effectively monitor the 3D spatial distribution of many different bacteria in oral biofilms. After obtaining high-resolution images, image processing and data analysis can quantify the biomass of different targets in oral biofilms [95]. Dige et al. successfully applied FISH to analyze spatial relationships in dental plaque and the changes of specific microflora over time [64]. Welch et al. used CLASI-FISH to discover a distinctive, multigenic consortium in the microbiome of supragingival dental plaque, which consisted of a radially arranged, nine-taxon structure organized around cells of filamentous corynebacteria. The size of this consortium varies from ten to hundreds of microns and follows a trend that promotes the spatial distribution of its flora. It was found that anaerobic taxa tend to be located in the interior, while parthenogenic or exclusively aerobic bacteria tend to be distributed in the periphery. Based on this finding, they propose a new morphological model of the microbial community in dental plaque biofilms [121]. In summary, FISH can accomplish both the identification of individual strains and achieve dynamic monitoring of the colony system. By combining with CLSM, it is even possible to visualize information such as the location of a particular strain in the biofilm, which cannot be achieved by traditional detection methods.

### 3.3. Effectiveness Evaluation

The purpose of studying oral microorganisms is to maintain a healthy oral microecology, so scientists make efforts to establish a fairly complete effectiveness evaluation system. At present, the research topics related to using FISH as one of the detection methods mainly include molecules, oral materials, and the establishment of models and methods.

#### 3.3.1. Evaluation of the Effectiveness of Molecules

In the long history of people dealing with oral pathogens, researchers have found that some molecules can specifically inhibit some of these pathogens and promote the reproduction of probiotics. The first kind of molecules are part of our daily lives, and it is commonly known that some have medicinal properties, such as salicylic acid, podophyllotoxin, and vinblastine. Hannig et al. [99] suggested that gargling with some polyphenol beverages and ingesting related foods could reduce the adhesion of initial bacteria to dental enamel and might help prevent diseases caused by oral biofilm. Similarly, Hertel et al. [100] used FISH to observe the anti-acid effect of Inula viscosa and its effect on the initial formation of oral biofilm and found that flushing with Inula viscosa for more than 8h had an effect on bacterial colonization on the enamel surface, but no effect on the anti-acid performance of the biofilm. 

The second kind of molecule is related to the artificial synthesis and improvement of natural compounds. Lyu et al. [101] studied the effect of LCG-N25 (a new small molecule exploited from known inartificial lead compounds) on the constituent parts of multi-species biofilm by using species-specific FISH. Their results showed that with good anti-bacterial activity and low cytotoxicity, this molecule cut down the proportion of Streptococcus mutans, Streptococcus sanguinis and Streptococcus gordonii but did not induce drug resistance to cariogenic *S. mutans*, manifesting that LCG-N25 could be a hopeful adjuvant for the treatment of caries.

After exploring the role of individual molecules, our team identified that it was equally important to understand the effects of the interaction of different molecules on our results, so a new series of experiments was carried out. Cheng et al. [102] investigated the combined impact of arginine and fluoride on oral bacteria and spotted that arginine might increase the ecological benefit of fluoride by enhancing the alkali-producing bacteria in plaque biofilm, playing a synergistic role with fluoride in the control of caries. In addition to the interaction of medicinal molecules, the gain effects of medicinal molecules on functional but non-medicinal molecules were also previously studied by other scholars. Liu et al. [103] added dimethylaminododecyl methacrylate (DMADDM) to a root canal sealant called EndoREZ and detected the composition of multi-strain biofilm by FISH and RT-qPCR. It was unveiled that when the mass fraction of DMADDM increased to 5%, the cytotoxicity, apical sealing ability, antibacterial, and solubility of the sealant were considerably distinguished from the control group. Therefore, it was concluded that the EndoREZ sealant containing DMADDM could be used for clinical prevention and treatment of persistent periapical periodontitis.

In addition to FISH, other technologies are also widely utilized, such as using PCR to explore the effects of terminalia chebula extracted from ethylalcohol on *S. mutans* [122], using DNA microarray to study the effect of secondary carbon dioxide laser irradiation on *S. mutans* [123], and so on. However, because PCR can only provide data analysis and can not show the results more intuitively, and DNA microarray can only be used to study the species that have been identified, FISH is more advantageous in evaluating the effectiveness of molecules.

#### 3.3.2. Evaluation of the Bacteriostatic Effect of Oral Materials

The oral cavity is an essential human structure closely related to food nutrition absorption, gas exchange, and speech expression. Consequently, dental materials are required to have biosafety, biocompatibility, and biofunction. However, the biofilm formed by microbes attached to the material surface can lead to drug-resistant infection and the contamination of medical devices [124]. Accordingly, the scientific community has explored different means to make bacteriostatic dental materials. In the field of dental implant materials, for instance, the development of bacteria-driven mucosal inflammation and peri-implant inflammation may lead to the failure of treatment. Al-Ahmad et al. [104] used FISH to detect the thickness, surface coverage, and oral *Streptococcus* spp. content in the biofilm, revealing that the surface roughness of the zirconia surface of the oral implant with low roughness was similar to that of titanium surface in terms of initial bacterial adhesion or oral *Streptococcus* spp. content in the biofilm. Similarly, PCR was used to unveal how TiO2 in the form of nanotube inserted into GIC affect *S. mutans* [125], but it was limited to gene expression analysis. If FISH can be used in this study, researchers can further analyze the morphological changes and distribution of bacteria.

#### 3.3.3. Verification of the Establishment of Models and Methods

The model method is a bridge between scientific theory, an effective means of successful scientific research, and a carrier for the sublimation of creative thinking. The research community has been exploring this method and expanding the knowledge about it each day, shedding light on the crucial role it can play in modern medicine. Many important experimental results have been achieved via teaching models in the classroom, experimental animal models and cell models, bioreactors, computer-aided 3D modeling, and mathematical modeling. At present, the model research on oral diseases caused by oral microbes can be divided into in vivo and in vitro models. For in vivo models, more clinically significant indicators such as plaque index and alveolar bone resorption are often used to judge whether the model is successful or not. As for the in vitro models, especially when the results are verified by FISH, the bioreactor is widely used to simulate the physiological or pathological environment. Different models, including an oral biofilm model of dental pulp disease established on hydroxyapatite and dentin disc [105], a repeatable and easy-to-use model for cultivating oral multi-species biofilms in a flow chamber system [106], a model combining in vivo and in vitro oral biofilm growth [107], and an in vitro “submucosal” biofilm model for peri-implantitis [108] have been established by in situ identification using FISH. It is believed that more different ones will be developed in the future, bringing upgrades into the scientific community and strengthening clinic knowledge.

If the model method is an up-to-date summary of the major findings of systematic research, then the establishment of the new methodology represents the improvement or subversive creation of the existing methods. However, the improvement or innovation of oral microbiological research methods requires more convincing proof methods, e.g., conventional slide detection, immunological methods, and molecular biology techniques [126], which makes FISH stand out as a technology with reasonable specificity, fast detection, and strong visibility. For instance, a microscopic method for macroscopic non-invasive monitoring of oral biofilms was created by Karygianni to depict the spatial distribution of biofilms on the bovine enamel surface (BES), and used FISH to verify the effectiveness of this new method. In this specific study, microbes were stained using specific-FISH probes, and signals were captured by CLSM and monitored by Scan∧R [109]. Analogously, Jackson et al. [127] used RT-qPCR to assist validation and establish a three-dimensional oral tissue model to study HPV. This model is of vital essence for the prevention and treatment of HPV-related diseases. If FISH can be introduced in the subsequent use to analyze the distribution and dynamic changes of HPV, the research results will be more meaningful.

## 4. Discussion and Perspectives

In recent years, FISH has significantly impacted clinical and microbiological detection methods due to its high specificity and efficiency, making it one of the most promising methods for clinical microbiological detection. In addition, FISH is commonly used in daily life experiments and may become a standardized experimental method in future research. Multiple optimizing standpoints of FISH have spawned numerous variants. Based on four fundamental procedures, innovations in probe design have been successfully put into practice, greatly expanding the applications of this methodology. Regarding potential applications, the specific fields are featured with diagnosis assistance, oral microecology research, and effectiveness evaluation, manifesting the great applicability of FISH in oral microbiology. 

Although the multimolecular affinity and good biocompatibility of FISH make it capable of providing a new and effective method for detecting oral microorganisms, the method still has its shortcomings, especially when it comes to potential interference with false-negative results due to signal loss. Albeit multiple variants of FISH have been improved from several perspectives, the ensuing more complex procedures and higher demands on experimental manipulation and equipment limit their practical applications. For basic FISH, the probes present the problem of the inability to achieve 100% hybridization rate, especially with cDNA probes which are complementary to mRNA. The detection of mRNA is mainly used for transcriptomic analysis but the level of mRNA does not fully reflect the level of expressed proteins. In addition, a lack of standardized processes for the analysis of FISH results exists, which leads to the need for experienced analysts to ensure relatively correct judgment, which undoubtedly brings extra training costs. At present, some disease sites only use detection technology to confirm the existence of oral microorganisms, and most of the research on FISH detection of oral microorganisms focuses on the impact on oral diseases, lacking research on other diseases that have been proved to be related to oral microorganisms. Therefore, FISH and its variants can be used in the future to analyze the community structure, dynamic evolution process, interaction with other cells and the relationship with disease occurrence and development of oral microorganisms in these lesions, so as to help doctors and researchers better develop and use drugs and other interventions.

With the advent of the era of precision medicine, microbiome and its precise regulation are exceedingly important in the research of microbial-related diseases. Thereby, the common application with other technologies and the innovation of FISH itself has become a general trend within the scholarly community, and the emergence of CLASI-FISH and HiPR-FISH are relevant examples. The emergence of these new technologies enables us to understand the interaction among genes, metabolites, and signal pathways in the oral microbial community from an ecological point of view rather than the previous relatively single interpretation [128].

Altogether, the future direction of FISH can be divided into two categories: (1) The improvement of FISH itself. For example, the further improvement of the accuracy of probes, the way to expand the variety and total number of probes while reducing the interference effect, and cost reduction. (2) The cooperative application of FISH and other technologies. For instance, to develop a full-process automated FISH experimental device by combining the image recognition technology of machine learning and deep learning and using FISH to analyze the three-dimensional models of oral microorganisms with real-time and dynamic monitoring. Hypothetically, if the ideas get actualized and widely accepted in the future, problems that need to be discussed from the perspective of “microorganism-host-system” can be studied more conveniently, quickly, and accurately. For instance, the establishment of a dynamic competitive ecological model of dominant bacteria in oral cavity, study on the evolution of pathogenic bacteria in hotbeds such as dental plaque, and the potential ways of oral pathogens to invade other parts of the body. Based on the dynamic three-dimensional database of oral microecology obtained by FISH, we can further explore the fundamental relationship between oral microorganisms and tumors, atherosclerosis, Alzheimer’s disease and other diseases, so as to develop monitoring equipment for oral microbial detection based on FISH probes to cooperate with other auxiliary tests to predict the risk of disease, determine the degree of disease development and judge the prognosis. This flourishing research field will scale to new heights based on these achievements.

## Figures and Tables

**Figure 1 pathogens-11-01450-f001:**
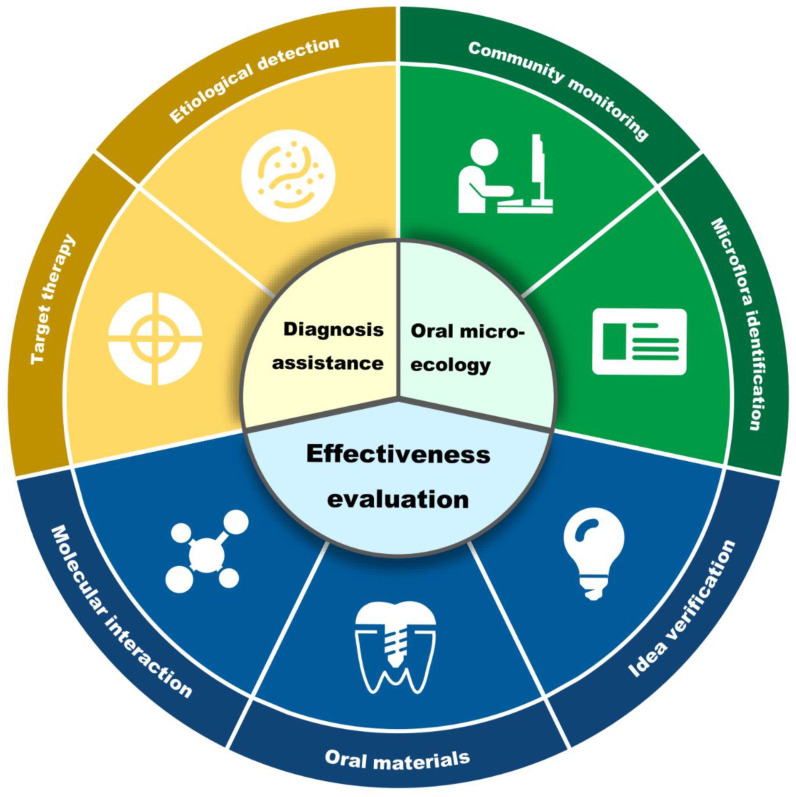
Applications of FISH in oral microbial detection. This figure summarizes the applications of FISH in oral microbiology into three categories: diagnosis assistance, oral microecology, and effectiveness evaluation. The specific applications consist of target therapy, etiological detection, community monitoring, microflora identification, molecular interaction, oral materials, and idea verification.

**Table 1 pathogens-11-01450-t001:** Comparison between FISH and common microbial detection techniques.

Method	Resolution	Culture Reliance	Unsearched Species	Quantification	Microbiota Analysis	Oral Microbiology Applications	Specialty	Reference
Fluorescence in situ hybridization (FISH)	Genus to Species	No	Detectable	Semi-quantification (affected by hybridizing rate)	Digital image analysis provides spatial resolution	Experimental study Rapid Clinical detection and diagnosis	Intuitive imaging and spectral quantitative positioning analysis	[13,33,34,36]
Polymerase chain reaction (PCR)	Subspecies to strain	Isolate nucleic acids from pure cultured bacterial cells	Detectable	Absolute quantification without calibrated standard or highly efficient amplification	Detect and quantify the community compositions	Experimental study Clinical detection and diagnosis	Some subtypes can identify microbes by themselves or be as a sample preparing process for other identification methods	[66,67,68,69]
DNA Microarray	Species	No	Only detectable to previously identified species	Quantitative detection of multiple bacteria	Monitor the changes of multiple species, capture the major species	Experimental study Epidemiologic investigation	Based on the pre-constructed microarray chip	[70,71]
Next-generation sequencing of 16s rRNA	Subspecies	No	Detectable	Unable	Assess taxonomic diversity of microbiota	Experimental study Clinical treatment evaluation Develop the databases of bacterial genomes	Mainly applied for community composition, evolutionary relationships, and diversity	[71,72,73,74]
Next-generation sequencing of whole-genome sequencing	Strain	No	Detectable	Absolute quantification	Unable	Develop the databases of bacterial genomes	Massive DNA sequencing with a high throughput but high cost	[73,75,76]
Restriction fragment length polymorphism (RFLP)	Species of several genera	Same as PCR	Detectable	Unable	Assess the diversity of complex microbiota and rapidly compare the structure from different environments	Oral microbiota analysis in smaller laboratories	An inexpensive but complex identification via obtained RFLP patterns	[77,78]
Denaturing gradient gel electrophoresis (DGGE)	Species	No, based on 16S rDNA amplified by PCR	Detectable	Semi-quantitative accompanied by real-time PCR	Generate 16S rDNA band patterns as species fingerprints	Experimental study Clinical treatment evaluation	isolate at least 10 different bacteria in each sample	[79,80]
Matrix-assisted laser desorption/ionization-Time-of-flight mass spectrometry (MALDI-TOF MS)	Genus to Species	Directly identified by protein or nucleic acid from samples, but the accuracy is higher after separation and purification	detectable	Relative quantification of targeted biomarkers	Mass spectral patterns represent bacterial distribution and relative abundance	Experimental study Rapid clinical detection and diagnosis	Significantly decreases the processing time, but requires expensive instrumentation fail in the identification of mixed infections	[81,82,83]
Enzyme-linked immunosorbent assay (ELISA)	Species	No	Only detectable to targeted species corresponding to infection	The level of inflammatory cytokines and immunoglobulins correlate to bacterial load	Unable	Clinical infection diagnostic examinations	Commonly adjunctive tool in clinical practice	[60,65,84]
Single-cell Raman spectra (SCRS)	Species	No	Detectable	Quantitative detection of individual droplets	Explore the mechanism of individual microorganisms, but unable to discern bacteria in complex environments	Single-cell investigation Rapid identification and classification	Label-free and non-destructive, but low throughput	[58,85,86]

**Table 2 pathogens-11-01450-t002:** Research and application of FISH in oral microbial detection.

Areas	Aspects			Ref
Diagnosis assistance	Etiological detection	Human immunodeficiency virus		[90]
		Epstein-Barr virus		
		Influenza virus		
		Avian infectious bronchitis virus		
		SARS-CoV-2		
		Treponema pallidum		
	Target therapy	Oral lichen planus	*Prevotella melaninogenica*	[91,92]
			Capnocytophaga	
			Gemella	
			Escherichia-Shigella	
			Megasphaera	
			Carnobacteriaceae	
			Flavobacteriaceae	
		Halitosis	Eubacteria	[93]
			Fusobacterium nucleatum	
			*Streptococcus* spp.	
Oral microecology	Oral microflora identification	In situ identification		[93,94]
		Biomass quantification		[95]
		3D spatial distribution		[89,94]
		Cobacteria in infection		[96]
	Community monitoring	Process of invasion, colonization and transmission		[59,97]
		Interactions between colonies		[98]
		Microbial identity & location map		[89]
		Effects of external factors		[15]
		Occurrence and progression of the disease		[98]
Effectiveness evaluation	Molecules	Natural molecules	Polyphenol beverages: reduce the adhesion of initial bacteria	[99]
			Inula viscosa: anti-acid and initial biofilm formation effect	[100]
		Artificial compounds	LCG-N25: adjuvant for the treatment of caries	[101]
		Combined molecules	Arginine + fluoride: synergistic control of dental caries	[102]
			DMADDM + EndoREZ: clinical treatment of periapical periodontitis	[103]
	Oral materials	Dental implant materials	Bacteriostatic effect: oral streptococcus content in the biofilm	[104]
	The establishment of models and methods	Oral biofilm model of dental pulp disease established on hydroxyapatite and dentin disc		[105]
		Repeatable and easy-to-use model for cultivating oral multi-species biofilms in a flow chamber system		[106]
		In vivo and in vitro oral biofilm growth model		[107]
		In vitro “submucosal” biofilm model for peri-implantitis		[108]
		FISH: a microscopic method for macroscopic non-invasive monitoring of oral biofilms		[109]

## Data Availability

Not applicable.

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
