# Peer review of "Application of Fluorescence In Situ Hybridization (FISH) in Oral Microbial Detection"

_pathogens, 2022, doi:10.3390/pathogens11121450_

Round 1
Reviewer 1 Report
This is a comprehensive review of FISH-based techniques in the field of oral microbiology and beyond. It is relatively well organised - however here are some comments that may improve the manuscript.
1) The section on diagnosis assistance gives examples of cases where FISH may be used to assist in diagnosis. The reviewer is not aware of FISH being used more widely to this end anywhere in the world - are the authors aware of examples where this is the case? Whilst the other sections deal with current research and applications of FISH - this section is discussing potential use cases rather than current usage. I suggest the authors make that clear.
2) Some of the points made in the Perspective sections are repetitive from previous sections. Suggest the authors discuss the techniques & applications only in the previous sections and provide critique in the final section if possible or remove the repetitive points in the last section.
Reviewer 2 Report
Comments are in the pdf. Major revisions required

Round 2
Reviewer 2 Report
Revisions are to my satisfaction